# Geological evidence for extensive basin ejecta as plains terrains in the Moon's South Polar Region

Le Qiao [1] ✉, Luyuan Xu [2], James W. Head [3], Jian Chen [1], Yuzheng Zhang[1], Bo Li[1] & Zongcheng Ling [1]

Water ice and other volatiles that accumulated in the Moon's polar regions are among the top priority targets for lunar exploration, due to their significances in both lunar geology and extraterrestrial resource utilization. Locating suitable landing sites and determining the provenance of sampled/measured surface materials are critical for future landed missions. Here, we map over 800 sites of plains terrains in the Moon's south polar region, with a total surface area of ~46,000 km². Orbital measurements and analog studies show that most of these plains have apparently higher albedo and lower iron content than volcanic mare plains, suggesting an origin of ejecta-induced debris flows from distant impact craters, especially from the Schrödinger basin. Our findings suggest that the entire lunar south polar region probably have experienced contributions from distant basin materials. We recommend these plains as priority landing sites for future exploration of lunar polar volatiles and early bombardment history.

Water ice and other frozen volatiles are thought to have accumulated in the permanently shadowed regions in polar regions of the Moon due to the extremely cold environments over geologically long periods of time[1–4]. Some of these volatiles may have been delivered to the lunar surface since the very early stages of solar system history, and they are therefore of scientific significances for solar system studies, e.g., the source and migration of solar system volatiles, and formation and evolution of planets[5,6]. Moreover, these ice deposits can be potentially harvested as a resource to enable sustained human exploration to the Moon and beyond in space[7]. In addition, the lunar samples collected in this context provide fundamental information about the history of the Moon and the provenance (origin) of both the samples[8] and the associated volatiles[9]. As a consequence, these polar volatiles have emerged as high priority targets for lunar science and exploration[5,10]. In the coming decade, dozens of orbital and landed missions are aiming to explore and assess lunar polar volatiles, especially in the southern polar region. For instance, NASA's Artemis III mission (currently scheduled for launch in the mid-2020's) is planned to send astronauts to explore the southern polar region of the Moon[11,12]. Moreover, China and Russia are jointly proposing the International Lunar Research Station, a long-term research station near the lunar south pole[13]. In addition to frozen volatiles, the south polar region of the Moon is also adjacent to the South Pole-Aitken (SPA) basin, the largest and oldest preserved lunar impact basin[14,15], providing an unparalleled opportunity to study deep-seated lunar materials[16] and the early bombardment history of the Moon.

The design and implementation of these lunar south polar missions all require precursor scientific understanding of the lunar southern polar region, in particular the geological characteristics and evolution of the candidate landing sites. These geological investigations will provide critical mission support, including landing and roving safety, provenance of the in-situ measured and returned samples, and modification processes experienced by the polar volatile deposits, support well demonstrated by the very successful Apollo missions[17].

As shown in the imaging and topographical maps (Fig. 1), also in recent[8] and earlier[17,18] geological mapping, the southern polar region of

[1]Shandong Key Laboratory of Optical Astronomy and Solar-Terrestrial Environment, School of Space Science and Physics, Institute of Space Sciences, Shandong University, Weihai, Shandong 264209, China. [2]State Key Laboratory of Lunar and Planetary Sciences, Macau University of Science and Technology, Macau, China. [3]Department of Earth, Environmental and Planetary Sciences, Brown University, Providence, RI, USA. ✉e-mail: leqiao@sdu.edu.cn

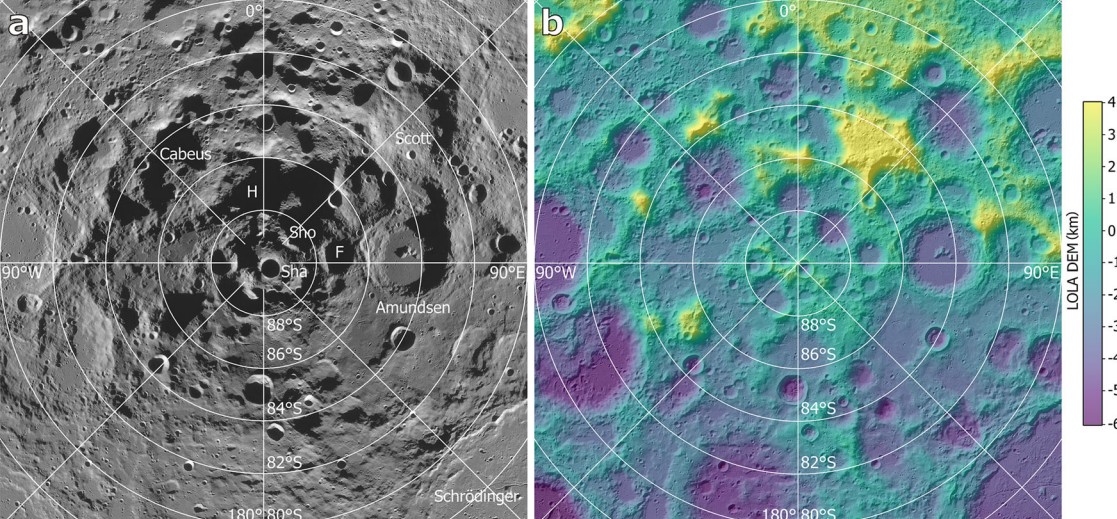

**Fig. 1 | Contextual maps of the southern polar region of the Moon. a** LRO (Lunar Reconnaissance Orbiter) WAC (Wide-Angle Camera) image mosaic (100 m/pixel) with minimal shadows (summer mosaic)[58]. **b** LOLA (Lunar Orbiter Laser Altimeter) gridded topography (20 m/pixel)[56] overlaid on a LOLA-derived shaded relief map (315° azimuth angle and 10° altitude angle). Major impact craters are labeled in (**a**): F, Faustini; H, Haworth; Sha, Shackleton; Sho, Shoemaker. All the maps for the lunar south polar region in this paper are projected into a stereographic projection centered at the south pole.

the Moon lacks the extensive mare plains like those in the landing areas of many previous missions that provide flat terrains to ensure safe landing. Instead, the southern polar region surface is dominated by feldspathic highland terrains overprinted by abundant impact craters of a wide range of dimensions (up to ~300 km in diameter) and ages (from pre-Nectarian to Copernican[8,18]), and ejecta from these craters and basins[19], a terrain resembling the Apollo 16 site and other heavily cratered terrain on the Moon (e.g., the Feldspathic Highlands Terrane[20]). This results in the very rugged long-wavelength topography (Fig. 1; median roughness (root mean square height) ~66 m at 1.82 km baseline) and a complex geological history. Locating suitable surface landing sites is critical for mission design and planning, as well as scientific understanding of the mission results.

Preliminary morphological examination, and previous geological mapping efforts[8,21–23] identify many small (relative to typical lunar mare plains) plains terrains in the southern polar region of the Moon (Fig. 1), especially on the floors of many craters, for instance, the Amundsen crater (Fig. 2a, b). This configuration is morphologically similar to many plains-filled craters in the equatorial regions of the Moon[23,24], such as the mare-filled Colombo crater (~79 km diameter, centered at 15°S, 46°E; Fig. 2c) and the light plain-filled Plummer crater (~73 km diameter, 25°S, 155°W; Fig. 2d).

Compared with the rugged topography in most areas of the lunar south polar region (median slope ~8° at 60 m baseline and median roughness ~25 m at 660 m baseline; Fig. 1 and Supplementary Fig. 1b), these polar plains are topographically very flat (median slope ~3° at 60 m baseline and median roughness ~7 m at 660 m baseline; Fig. 2b and Supplementary Fig. 1b), serving as attractive exploration sites for surface landing and roving lunar polar missions. However, a set of fundamental questions remain to be further investigated about these polar plains terrains: (1) What is their spatial distribution across the southern polar region of the Moon? (2) What are their detailed geological characteristics, including topography, morphology, composition, and chronology? Are they similar to or different from the well-known mare basaltic plains? (3) What is their origin and emplacement mechanism? Are they also volcanic in origin as are many shallowly buried cryptomaria, or basaltic mare plains on the Moon with different mineralogy and higher albedo? (4) Are these polar plains promising sites for lunar polar volatile exploration? What can we learn from these polar plains about the stratigraphy, emplacement, and age of volatile

deposits at lunar southern polar region? These questions are critical for the design and planning of future lunar polar exploration efforts.

In this contribution, we focus on the plains terrains in the lunar polar region southward of 80°S latitude. We employ the latest high-resolution and high-precision topographic and other data sets to access the spatial distribution and detailed characteristics of these polar plains. These analyses are then combined with analog studies to investigate their formation mechanism and potential source regions. Finally, we discuss the science and exploration value of these polar plains, including both volatile prospecting and other scientific opportunities.

## Results
### Distribution of plains terrains in the lunar south polar region
Using high-resolution LOLA topography and derived maps, we undertook a thorough search for, and mapping of, plains terrains in the southern polar region of the Moon. We identified 873 sites of plains terrains with surface areas ≥1 km² (Fig. 3; limited by the 20–1000 m pixel sizes of the various data sets, see Methods) in the region shown in Fig. 1. Smaller plains units were also found but are not cataloged and characterized in this work. The surface area of these mapped plains ranges over nearly four orders of magnitude, up to over 8000 km², with a median value of 9.3 km² (Supplementary Fig. 2). Areally extensive plains occurrences include those emplaced on the floor of large craters, e.g., Schrödinger (8138 km²), Drygalski (2584 km²), and Amundsen (2322 km²). The histogram of surface areas is characterized by a leptokurtic distribution, with a positive skewness toward larger areas (Supplementary Fig. 2). Smaller plains are much more common than larger ones: ~53% of the plains are smaller than 10 km² and ~86% are smaller than 50 km². All the mapped plains have a total surface area of about 46,700 km², constituting ~12.8% of the entire study region (Fig. 1).

A significant part (~57%) of these plains are located on the floors within craters, especially for many relatively larger-area plains occurrences (Fig. 3). These crater-interior plains are generally characterized by very low slopes (typically <2°), smooth surface textures (topographical roughness typically <10 m at 660 m baseline), and distinct contacts with the more rugged and steep crater walls (Fig. 4a). They usually have an approximately circular outline defined by the sloped wall terrains. In addition, we also observed many plains terrains (~43%) at locations outside distinct impact craters (Fig. 3). These inter-crater

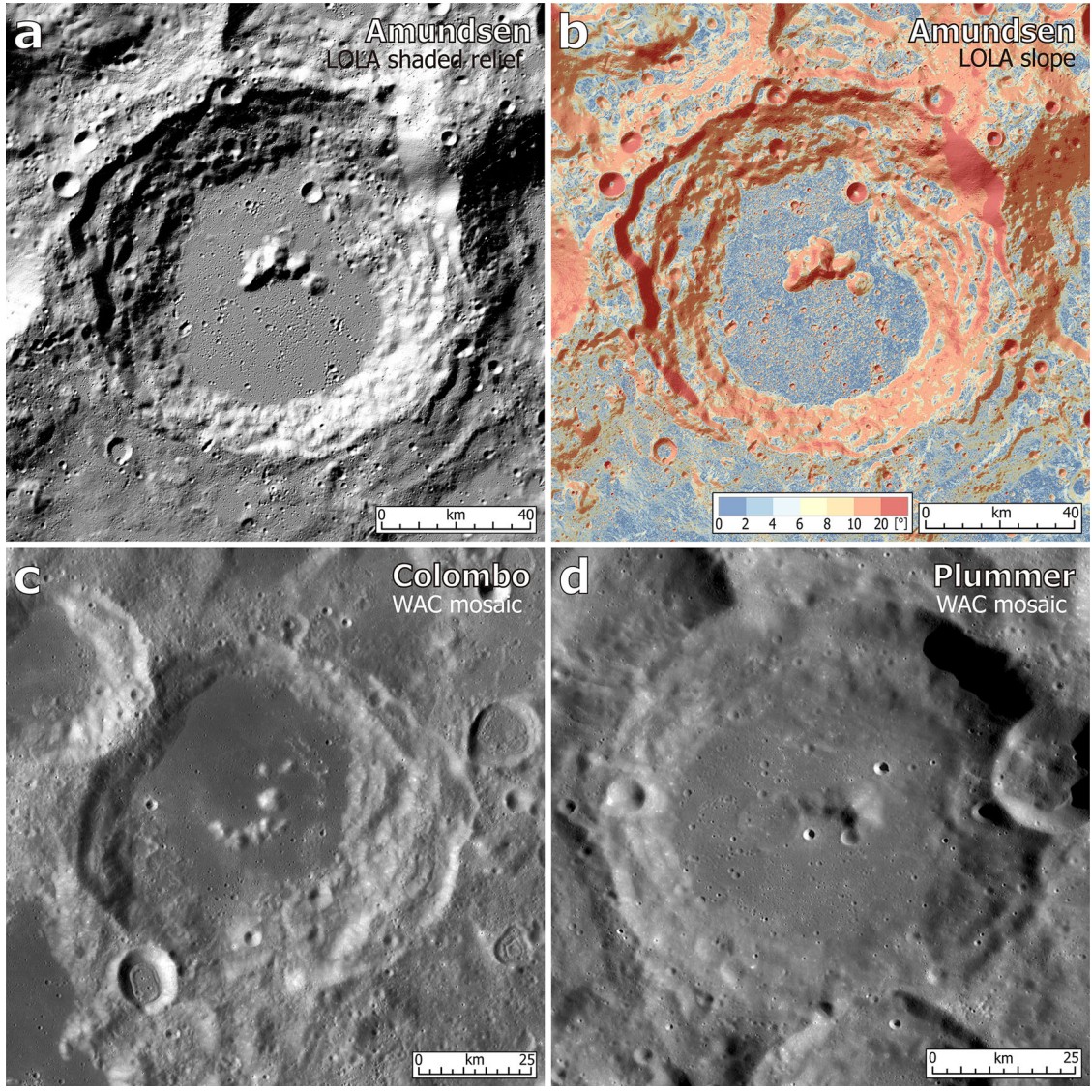

**Fig. 2 | Representative plains terrains in the south polar and equatorial regions of the Moon.** LOLA (Lunar Orbiter Laser Altimeter) (**a**) shaded relief and (**b**) slope maps of plains on the floor of the polar Amundsen crater (103 km diameter, centered at 84.4°S, 83.1°E). LRO (Lunar Reconnaissance Orbiter) WAC (Wide-Angle Camera) maps of plain-filled craters in the equatorial (**c**) Colombo (mare-filled, 79 km diameter, 15.3°S, 46.0°E) and (**d**) Plummer (light plain-fillled, 73 km diameter, 24.6°S, 154.9°W) craters. Maps of the Colombo and Plummer craters are both cropped from the WAC global morphologic mosaic and projected into a plate carrée projection.

plains have slightly elevated topographical slopes (though mostly <5°), while they still retain smooth surface textures (Fig. 4b), similar to those of crater-interior plains. Many of these inter-crater plains appear to be superposed on the underlying sloped terrains, often with elongated outlines. In these cases, their spatial boundaries from adjacent terrains are often more gradational.

We note that a portion of the polar plains that we mapped, were also mapped as light plains by ref. 23 (Fig. 3). Whereas our mapping effort has identified significantly more plains than previous work: only 95 sites of plains ≥1 km² were mapped by ref. 23 in the lunar south polar region, with a total surface area of ~29,700 km². This discrepancy is mainly attributed to the new LOLA data sets with much higher resolution (20 m/pixel) and better spatial coverage employed in this work, compared with the 100 m/pixel LROC WAC data used by ref. 23, with considerable shadowed areas, especially for regions south of 85°S latitude (Fig. 1a). In addition, we mapped these plains at various scales from 1:200,000 to 1:400,000, compared with the 1:300,000 scale by ref. 23.

## Geological characteristics

We first referred to the gridded LOLA data and derived maps (Figs. 1b and 2b and Supplementary Fig. 3) to examine the topographic characteristics of the plains terrains in the lunar south polar region. We plotted the histograms of the elevation, slope, and roughness values for the polar plains and the entire south polar region (Supplementary Fig. 1). The plains terrains are characterized by a wide range of topographic elevations from about −7 km to about 6.7 km relative to the mean lunar radius (1737.4 km), while the elevation of the majority (~75%) of these plains range from −5 km to −2 km (Supplementary Fig. 1a), with peak values around −4.7 km. Compared to the entire southern polar region (peak values around −2.3 km), the plains generally lie at lower elevations, as in the cases of those on the impact crater floors. The histogram plot shows an approximately unimodal distribution pattern for the entire southern polar region, while a number of peaks are observed for the mapped plains terrains. This comparison suggests that these plains are likely to have been emplaced after the formation of the primordial highland terrain in

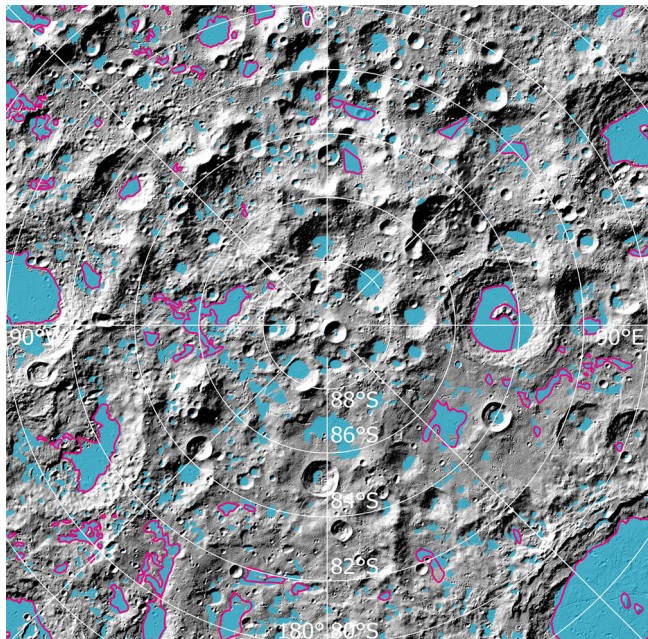

**Fig. 3 | The spatial distribution of plains terrains in the lunar south polar region.** Mapping results of this work are shown as turquoise patches and those of ref. 23 are shown as pink polygons.

the southern lunar polar region and that their topography is superposed on that of the underlying terrain. The slope-frequency distribution of both the plains terrain and the entire southern polar region exhibits a distinctive unimodal structure, with a downward roll-over at slopes near zero, a common slope signature for natural surfaces. Expectedly, the polar plains are characterized by very flat slopes, with ~74% of the surface flatter than ~5° and most-frequent slopes around 2°, compared with ~4° for the entire southern polar region (Supplementary Fig. 1b). In addition, these plains terrains have lower roughness values at all three baselines (Supplementary Figs. 1c, d, and 3).

We then investigated the surface optical albedo and iron contents[25–27] of the mapped polar plains, and their comparison with those of other typical lunar terrains. In the histogram plots, we observed that the polar plains have nearly identical 1064 nm albedo and iron contents as those of the entire southern polar region (excluding the mapped plains), with median albedo around 0.32 and FeO abundance around 7.2 wt% (Fig. 5). The global light plains mapped by ref. 23 and the entire Moon also have similar most-frequent values of LOLA albedo and FeO content, while they also contain a considerable portion with lower albedo and higher iron content. (Note that the small discrepancy in the most-frequent FeO abundance values between the south polar region and the global Moon (median value: 7.2% vs. 6.5%; Fig. 5b) are likely attributed to the usage of different data sets (Lunar Prospector vs. Kaguya+LRO), as no such discrepancy is observed if the same Lunar Prospector data alone is analyzed (Supplementary Fig. 4).) The maria regions, however, display lower albedo (median value ~0.18) and elevated FeO abundance (median value ~16 wt%), distinctly different from the polar plains terrains (Fig. 5).

Finally, we conducted a systematic survey of the surface geomorphology and textures of the polar plains mapped in this work. We identified two major types of morphologies for these plains terrains. The first type is represented by those emplaced on the floor of the Amundsen crater (Fig. 4a). These Amundsen-type plains terrains are characterized by very low slopes and show flat, smooth surface textures, with little topographic relief. The major features superposed on these plains are many small craters, including both circular primary craters and elongated or irregular secondary craters and clusters.

Many of these plains occur on the floors of larger craters, while some are emplaced on terrains outside craters, and usually have slightly steeper slopes (e.g., Fig. 4b).

Another type of plains is represented by those occurring on the floor of Schrödinger basin (Fig. 4c). These Schrödinger-type plains are also generally flat, but their surface is populated by abundant small mounds and hummocky features. A network of fractures and graben has been identified by many previous studies on the Schrödinger basin floor[28]. We find that such fractures are uniquely observed in the plains within Schrödinger basin, and not in plains elsewhere in the southern polar region. The Schrödinger fractures are likely to have formed by dike propagation-induced shallow magmatic intrusion[29], and are not genetically related to the plains terrains, except due to the emplacement of some dark-halo pyroclastic deposits and small mare patches. As a result, these fractures/grabens are not designated among the shared properties of Schrödinger-type plains. On the basis of this classification scheme, we assessed the morphological type of each plain terrain and mapped its distribution (Supplementary Fig. 5). We found that these polar plains are dominantly Amundsen-type and only 9 occurrences of Schrödinger-type plains are identified (Supplementary Fig. 6a–i). All these Schrödinger-type plains are emplaced within impact craters, and none are observed outside craters. Except for the Schrödinger basin (312 km diameter), all the other host craters are much smaller, including Shackleton crater (21 km diameter) at the lunar south pole (Supplementary Fig. 6g).

## Origins and source regions

On the surface of terrestrial planetary bodies, the presence of plains terrains generally indicates the formation of a gravitational equipotential surface in the geological past, which typically involves the emplacement of fluidized materials, for instance, fluvial sediments on Earth and Mars. Considering the unique geological and environmental conditions of the lunar surface, especially the nearly anhydrous and vacuum conditions, many plain-formation geological process involving water and volatiles can be excluded and the possible candidates mainly include volcanic and impact-related processes. Our optical albedo and iron abundance measurements show clearly that these polar plains are apparently different from volcanic mare deposits (Fig. 5). This comparison indicates that these polar plains are unlikely to be volcanic basaltic mare plains. We also examine the surface albedo, morphology, and composition of the mapped plains in detail (see Methods) and do not observe any dark-halo impact craters (consistent with ref. 30) or mafic-rich regolith signature (Supplementary Fig. 7), showing that these polar plains are not underlain by basaltic materials (cryptomaria). (It should be noted that it is challenging to identify surface albedo variations in the polar regions using typical conventional optical images due to considerable shadows and typically very low Sun illumination.) This observation is also consistent with the deficiency of volcanic deposits detected in the polar regions of the Moon[8,31,32]. Impact cratering is known to produce various kind of materials and deposits, and some are able to flow during their emplacement. The most common impact-derived fluidized materials include impact melt flows[33,34] and impact debris flows consisting of pre-impact substrate and ejecta materials[35–37]. How to distinguish between the two kinds of impact materials for the nature and origin of our mapped polar plains terrains? We address this issue through geomorphology and analog studies of impact-derived plains terrains elsewhere on the Moon.

We first focus on the more prevalent Amundsen-type plains terrains with very smooth surface textures. Amundsen is a typical lunar complex crater with a diameter of 103 km and lies ~170 km from the south pole. We undertook a survey of lunar impact craters with sizes (90–110 km diameter) similar to Amundsen crater ($n = 187$ according to the catalog of ref. 38) and containing interior plains terrains. We found similar smooth plains on the floors of a number of impact

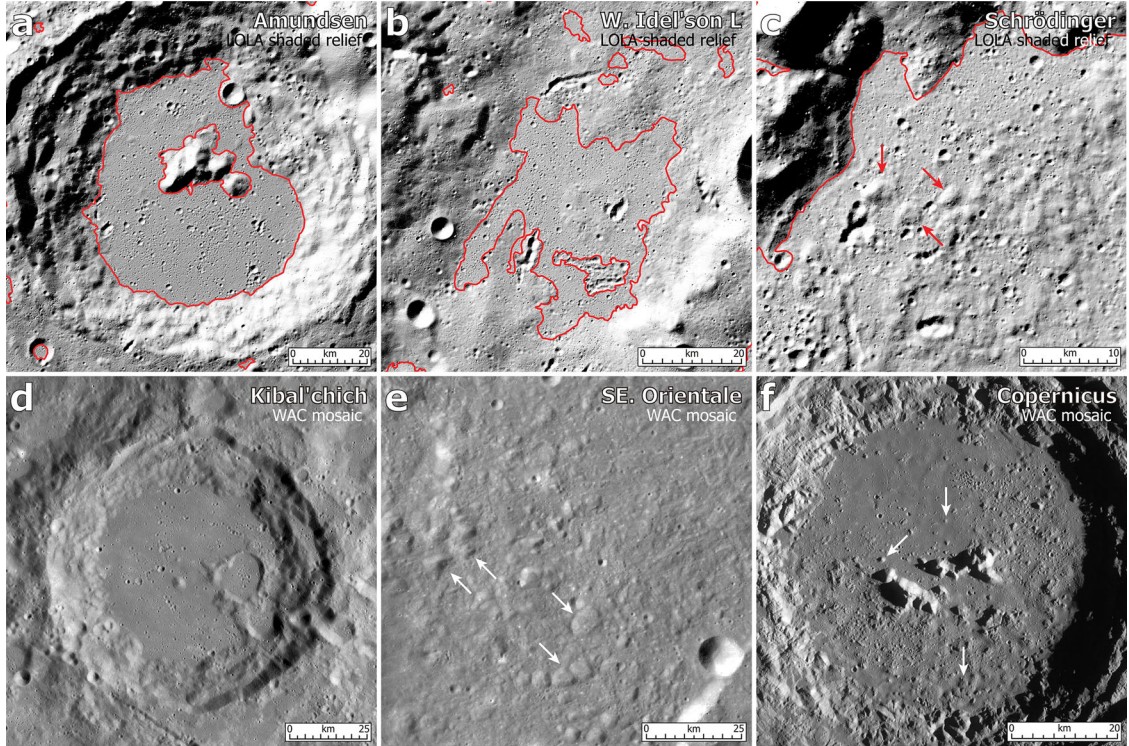

**Fig. 4 | Examples of lunar polar plains terrains and their analogs.** LOLA (Lunar Orbiter Laser Altimeter) shaded relief maps of polar plains (red polygons) (**a**) on the floor of the Amundsen crater (centered at 84.6°S, 87.6°E), (**b**) on terrains western outside of the Idel'son L crater (85.3°S, 128.5°E), and (**c**) on the southern floor of the Schrödinger basin (78.6°S, 125.1°E, mounds pointed out by red arrows). LRO (Lunar Reconnaissance Orbiter) WAC (Wide-Angle Camera) maps of analogs for lunar polar plains (**d**) on the floor of Kibal'chich crater (2.8°N, 147.1°W) with smooth surface textures, (**e**) on the southeastern Montes Rook Formation of the Orientale basin (25°S, 83°W) consisting of km-scale blocks (white arrows) surrounded by flat-lying materials, and (**f**) on the floor of Copernicus crater (9.7°N, 20.1°W) with characteristic mound features (white arrows). Maps of the Kibal'chich, Orientale, and Copernicus regions are cropped from the WAC global mosaic, respectively, and projected into a plate carrée projection.

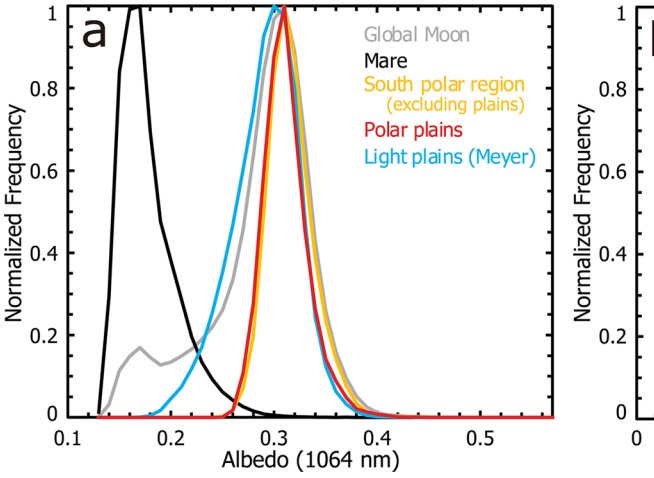

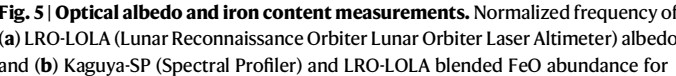

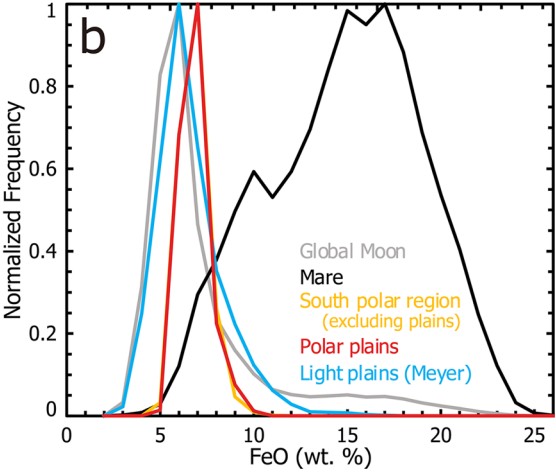

**Fig. 5 | Optical albedo and iron content measurements.** Normalized frequency of (**a**) LRO-LOLA (Lunar Reconnaissance Orbiter Lunar Orbiter Laser Altimeter) albedo and (**b**) Kaguya-SP (Spectral Profiler) and LRO-LOLA blended FeO abundance for the polar plains mapped in this work and their comparison with those of the global Moon, lunar maria, the entire southern polar region (excluding the mapped plains), and light plains mapped by ref. 23.

craters, for instance, Parenago (26°N, 109°W, 95 km diameter) and Kibal'chich (3°N, 147°W, 92 km; Fig. 4d). The plains terrain on the floor of Kibal'chich have been mapped previously by ref. 23 as light plains (Supplementary Fig. 8). The spatial distribution of the Kibal'chich plains and other adjacent plains demonstrate that they are radial to the ~930 km-diameter Orientale basin (Supplementary Fig. 8), supporting the interpretation of ref. 23 that they may originate from fluidized basin ejecta materials from the Orientale basin-formation impact.

On the basis of these analog and process studies, we suggest that Amundsen-type polar plains are likely to be ejecta deposits from distant impact craters or basins. It should be acknowledged that impact melt materials could have been produced during the impact cratering process and emplaced on the primitive crater floor, which would have been buried by and mixed with subsequent ejecta materials. These melt materials can be potentially sampled/measured by future landed missions to date the host crater and been compared

with previous crater population dating results[8]. We will investigate the potential source craters of these plains terrains in the latter part of this section.

We then turn to the Schrödinger-type plains terrains with characteristic mound features. As the second youngest (after Orientale[15]) impact basin on the Moon, the 315 km-diameter Schrödinger basin is regarded as one of the prototypes for studying large impact basins. We found that the light plains radial to the Orientale basin generally lack such mound features, for example, those emplaced within the 350-km-diameter Schiller-Zucchius (unofficial name) impact basin in the northeast of Zucchius crater (Supplementary Fig. 9), suggesting that the Schrödinger-type polar plains are probably not emplaced basin ejecta materials. In addition, we did not find craters or basins with a size comparable to Schrödinger that host similar mound-bearing plains terrains. Alternatively, these Schrödinger-type plains resembles many units of the Maunder (inside the Inner Rook Ring) and Montes Rook (between the Cordillera and Outer Rook Rings) Formation of the Orientale basin, which predominantly occur in topographically low regions, and contain mainly of flat-lying materials and km-scale blocks (Fig. 4e). The Maunder and Montes Rook Formation units have been generally interpreted as clast-rich melt materials produced during the basin-formation impact[39,40]. We also found that these Schrödinger-type plains are morphologically very similar to the floor terrains of some very young (Copernican) craters, for instance, Copernicus (10°N, 20°W, 96 km in diameter; Fig. 4f), Tycho (43°S, 11°W, 85 km), and Ohm (18°N, 114°W, 62 km). At Copernicus crater, abundant small mound features (up to ~1–2 km in size) are observed along the marginal portions of the crater floor, which were interpreted as broken, un-melted rock fragments embedded in the impact melt deposits[34], a common occurrence in young, unmodified lunar craters[33,41]. From these observations, we suggest that Schrödinger-type polar plains terrains are likely to represent local impact-related blocks and melt materials. Similar mound-bearing plains were not observed within impact craters of comparable size of Schrödinger because Schrödinger is among the youngest basins on the Moon and the primordial surface features of the impact melts within other older basins have been topographically degraded.

In summary, the two morphological types of polar plains terrains are interpreted to have different formation mechanisms and source regions: the smooth Amundsen-type plains are composed of mixtures of local substrate and ejecta materials from distant impact craters/ basins and the mound-bearing Schrödinger-type plains are local impact melt-related materials associated with the host craters. This interpretation is consistent with the stratigraphy of the host craters of these plains (Supplementary Fig. 5). Most of the Schrödinger-type plains terrains ($n = 9$; Supplementary Fig. 6a–i) occurs within relatively younger craters (compared to the host craters of Amundsen-type plains): two Eratosthenian-aged craters, five Imbrian-aged craters, and two Nectarian-aged craters (from the geological maps[8,18]). Their relatively younger ages make it possible to preserve the primary impact melt-related morphology. Most of the Amundsen-type plains occur in ancient (Nectarian and Pre-Nectarian) terrains, either crater floors or inter-crater terrains, as only huge impact basins (which are all very ancient: Imbrian or older) are able to produce sufficient ejecta materials to form the extensive plains terrains.

The geological provenance of the Schrödinger-type plains is local and related to the host impact crater, while the Amundsen-type plains consist of mixtures of both local substrate and ejecta materials delivered from distant impact craters/basins[35,36]. What are the source regions of the distal materials? Examination of the surface image and geological map[8,18] shows that no crater or basin is both larger and younger than the 315 km Imbrian-aged Schrödinger basin in the vicinity of the south polar region of the Moon. On the basis of this observation, we suspect that the Schrödinger basin may have been among the major sources of these fluidized ejecta materials.

To test this hypothesis, we conducted a case study of the Nectarian-aged Amundsen crater and calculated the theoretical ejecta thickness (Supplementary Table 1) at the center of the Amundsen floor (85°S, 88°E) from all lunar impact craters/basins ≥10 km in diameter (from the Robbins database[42]) using an empirical ejecta thickness model of Pike[43] with distance and deposition corrections (see methods in ref. 44 and references therein). We found that among the craters with a comparable or younger age of Amundsen (from the geological map of ref. 18), ejecta from Schrödinger basin is the thickest (~61 m), significantly thicker than the second largest ejecta contribution from Orientale basin (~13 m). The delivered ejecta materials from Schrödinger basin would have excavated and mixed with local materials with depth up to ~183 m (Supplementary Table 1), which together formed the ejecta deposit flows. No impact craters (up to ~2.9 km in diameter) superposed on the Amundsen floor plains are observed to excavate the underlying primitive crater floor substrates which are probably more cohesive than the surface fine-grained debris flow materials, suggesting the emplaced ejecta material are at least ~250 m thick (the maximum excavation depth of the superposed craters: 0.084*diameter[45]), close to the total thickness of local and foreign materials (Supplementary Table 1). We then calculated the areal abundance of plains terrains with distance from Schrödinger basin (Supplementary Fig. 10) as previous investigations of the Orientale basin found that the areal abundance of ejecta flows as light plains from the host basin reached ~20% as distance from the crater center increased to ~4 radii, and the areal abundance dropped below 10% beyond 4 radii[46]. We found that within one radius from the Schrödinger basin center, the areal abundance of plains terrains is very low (~5%; Supplementary Fig. 11), as the ejecta terrain is dominated by a hummocky, continuous ejecta deposit. The areal abundance increases to ~15% from one to two radii and remains at this high level out to a distance of four radii (Supplementary Fig. 11). We also investigated the areal abundance of light plains[22] moving away from Schrödinger in the northern direction (opposite to the southern polar region), and observed similar patterns, thought the areal abundance of light plains stars to drop at locations relatively closer to the basin rim (~3 radii), likely due to the coverage of some plains by mare deposits (Supplementary Figs. 12 and 13). This observation is generally consistent with the interpretation of Orientale ejecta light plains[46], supporting the hypothesis that Schrödinger basin is the major source crater for many smoother plains terrains in the lunar southern polar region. In addition, the maximum areal abundance of ejecta plains from Schrödinger (~15%) is slightly smaller than that of Orientale basin (~20%), indicating that other craters/basins might have also contributed considerable amounts of ejecta in the south polar region[19], especially the Orientale basin, as it is stratigraphically younger than Schrödinger basin and its ejecta should have been superposed on and mixed with the Schrödinger ejecta.

Finally, we investigated the surface model ages of the Amundsen floor plains and Schrödinger basin using superposed crater population analyses. Are their ages similar or different? We utilized the catalog of kilometer-scale impact craters from ref. 42 to analyze the crater size-frequency distribution (CSFD) characteristics of the Amundsen floor plains and the Schrödinger impact melt deposits. Possible secondary craters were excluded according to their spatial distribution and morphological patterns (Supplementary Fig. 14). The resultant cumulative CSFD plot shows that the crater density of the two areas is comparable in the 1–2 km diameter range. For craters >2 km in diameter, the Schrödinger floor has a relatively higher cumulative crater density than the Amundsen floor, but the difference is generally within statistical errors (gray bars). Chronological fitting of crater populations yields similar model ages (~3.8–3.9 Ga, with difference well within the uncertainties of the crater population dating technique[47]) for floor terrains of Amundsen and Schrödinger basins (Supplementary Fig. 14). In conclusion, the analysis results from ejecta thickness, areal abundance distribution, and crater population chronology all support

Schrödinger basin as a major source region for many smooth plains terrains at southern polar region of the Moon.

## Discussion

On the basis of its unexplored nature and distinctive scientific value, the lunar southern polar region and the cold-trapped and potentially abundant volatiles have become popular targets for exploration destinations on the Moon, in particular, for multiple planned landed missions. Due to the large-scale rugged topography of the lunar southern polar region (compared with mare plains that many previous missions landed on) and the unique environments for polar volatile emplacement and preservation, selecting suitable landing and exploration sites is critical for these scientific and volatile prospecting missions. Recent studies of the landing sites of lunar polar missions planned for the coming years have commonly focused on high-standing massifs, for instance, the western Nobile crater rim for NASA's Volatiles Investigating Polar Exploration Rover (https://www.nasa.gov/press-release/nasa-s-artemis-rover-to-land-near-nobile-region-of-moon-s-south-pole), the many candidate landing regions for NASA's Artemis III (https://www.nasa.gov/press-release/nasa-identifies-candidate-regions-for-landing-next-americans-on-moon), China's Chang'e-7[48] and beyond missions. The major advantage of these massif areas is the favorable solar illumination conditions, thus providing a relatively benign thermal environment and abundant solar energy for surface exploration activities. However, these high-standing massifs are still subject to considerable difficulties in both science and exploration. Topographic slopes for most portions of these massifs are very steep, commonly larger than 20° (ref. 49), very challenging for surface landing, rover mobility, and surface experiments. The summits of some massifs are relatively flatter, while their spatial extents are typically very small. For instance, the areas of slope less than 10° on the summit of Mons Malapert are less than 4 × 2 km in size. This is much smaller than the nominal landing ellipses for many lunar missions, such as the 15 × 30 km landing ellipse for the Russian Luna 25-Glob mission[50] and requires very high landing precision (Note that the landing ellipse for NASA's Artemis program is probably much smaller, at a scale of ~100 m[51].) In addition, frequent solar illumination would result in relatively warm surface temperatures that are not suitable for the long-term retention of water ice and other frozen volatiles. We expect that it will be very difficult to locate and harvest spatially extensive ice deposits on the surface of these massif terrains; the ice deposits are more likely to have accumulated either in sub-kilometer micro-cold traps or at some depth below the surface.

This study of the polar plains terrains provides an alternative perspective on the targets for future volatile prospecting missions to the Moon. The most apparent advantage of these polar plains is the presence of extensive areas of very flat topographic slopes, similar to many mare plains. For instance, the plains terrain on the floor of Amundsen crater is about 70 × 60 km in dimension and characterized by a mean slope of ~5°. This topographic configuration resembles the mare plain within the 168-km-diameter Von Kármán crater, the landing site of China's Chang'e-4 mission[52], and is able to accommodate the landing ellipses for many future lunar missions. Numerous regions of permanent shadow (PSR) have been identified on the Amundsen floor plains, including many small PSRs and some extensive PSRs near to the pole-facing walls (blue patches in Fig. 6). Thermophysical measurements show that some of these PSRs have maximum surface temperatures below 112 K (brown outlines in Fig. 6), providing thermal stability environments for surface water ice accumulation[53]. Previous studies have revealed several lines of evidence consistent with elevated hydrogen/water ice deposits on the Amundsen floor[4,22,54]. We suggest that many plains terrains on the Amundsen crater floor and elsewhere in the lunar south polar region would be attractive exploration targets for future ice-prospecting missions. Remarkably, at least 8 of the 13 Artemis III candidate landing regions contain the plains terrains mapped in this work (e.g.,

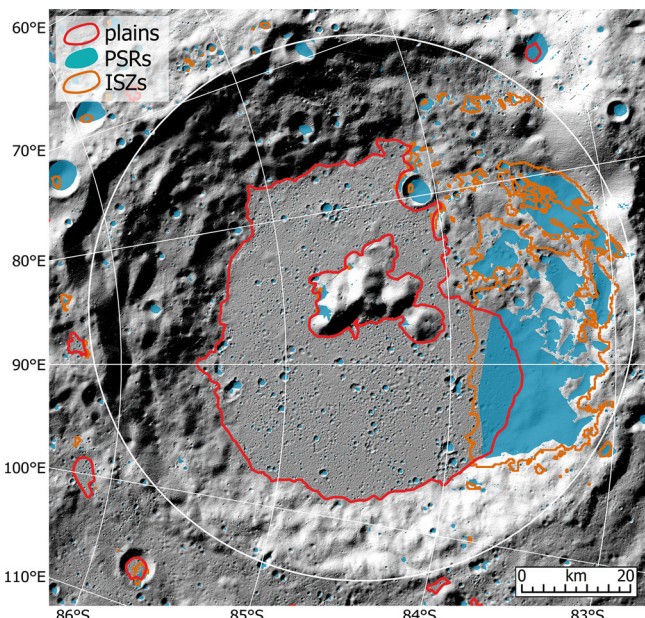

**Fig. 6 | LOLA (Lunar Orbiter Laser Altimeter) shaded relief map of the Amundsen crater floor.** The red polygons outline the plains terrains mapped in this work, blue patches are regions of permanent shadow (PSRs), brown outlines are ice stability zones (ISZs), and the white circles mark the crater rim positions.

the Peak Near Shackleton region, Supplementary Fig. 15), though these plains are typically very small in size (several km²) and occur at the marginal low-lying portion of these landing regions. In addition, we also observe many terrains that are not mapped as plains due to their relatively high topographic slopes, but they have smooth textures similar to many Amundsen-type plains, like those in the southern exterior of Amundsen crater (Fig. 3). This indicates that basin ejecta materials are probably much more extensive than the plains terrains mapped in this work and their elevated slopes are likely attributed to the underlying terrain.

While the solar illumination conditions in the mapped low-lying plains terrains is generally not as favorable as those of high-standing massifs, areas in these plains of substantial solar energy still exist. For example, the time-weighted illumination condition in the southern portion (left part of Fig. 4b) of the Amundsen crater floor plains can reach around 30%[22]. It is clear that the usage of advanced nuclear batteries would significantly aid future exploration of lunar polar volatiles.

In addition, these plains terrains in the southern polar region also provide a variety of scientific opportunities other than volatiles. Our mapping and source-tracking investigations show that most of these plains terrains (Amundsen-type) are composed of mixtures of local substrate and ejecta materials from distant impact craters/basins, especially the Schrödinger basin. Moreover, the accumulation of many plains materials within impact craters suggests that these depressions serve as containers for ejecta materials and foreign basin materials have probably contributed to the entire lunar south polar region. In-situ measurement or returned samples from these plain materials will provide exciting opportunities to study basin-formation impact process on the Moon. Isotopically-based dating analyses of these ejecta materials, either in-situ or measured in terrestrial laboratories, will obtain precise ages of the Schrödinger basin, which is vital for calibrating the lunar chronology function and studying the early bombardment history of the inner solar system.

Moreover, due to the inferred distant source for most of these polar plains terrains, special cautions are required when interpreting the measurements from these plains terrains. These plains materials

were emplaced after the formation of the host craters, and the primitive host crater floor terrains had been largely re-surfaced and buried. Crater population-derived model ages measured from these plains terrains probably represent those of the distance source craters, not the host craters[8,55]. The same issue applies to measurements of samples from these plains terrains.

## Methods

### Mapping of plains terrains
We identified and mapped plains terrains in the region within 10° latitude of the lunar south pole (i.e., the extent shown in Fig. 1) using maps of shaded relief, topographical slopes, and roughness calculated from high-resolution (20 m/pixel) altimetric gridded data from the Lunar Orbiter Laser Altimeter (LOLA)[56]. A shaded relief model of the lunar surface was produced from the LOLA topography data via mimicking the solar illumination effects (315° azimuth and 10° altitude angles). Such simulated shaded relief maps are commonly used for geomorphological analysis[22,57], especially for lunar polar regions where conventional optical images often contain a large percentage of shaded areas. A topographic slope map was calculated using a moving 3 × 3 pixel window (60 m baseline) to identify topographically flat terrains. In addition, topographic roughness maps were independently employed to map flat and smooth areas. In this work, the root mean square (RMS) of height/elevation over the analysis baseline was calculated as the topographical roughness. We derived roughness maps over a wide range of baselines and found that the combination of roughness at baselines of 660 m, 1020 m, and 1820 m is particularly useful for identifying plains terrains. Roughness maps at these three baselines were used to generate a pseudo-color image with 1820 m-baseline in the red channel, 1020 m-baseline in the green, and 660 m-baseline in the blue (Supplementary Fig. 3). As plains terrains should be smooth at all the three baselines, they, therefore, appear dark (i.e., low roughness values at each baseline) in the color composite roughness map[23]. We manually delineated the boundary of each plains terrain at various scales (from 1:200,000 to 1:400,000) from LOLA topography-derived maps.

### Geological characterization of plains terrains
We then analyzed the detailed geological characteristics of the mapped plains terrains using various types of orbital remote sensing data sets. LOLA gridded topography, and the derived slope and roughness maps are employed again to examine the topography of lunar polar plains occurrences and their comparison with the entire southern polar region. The LOLA shaded relief map, with assistance of the optical image mosaic with minimum shadows acquired by the Lunar Reconnaissance Orbiter (LRO) Wide-Angle Camera (WAC)[58], were used to analyze the detailed morphology of the mapped polar plains terrains. We compared their morphology with that of analog sites elsewhere on the Moon to explore their potential substrate properties and emplacement mechanisms. In addition, the catalog of kilometer-scale impact craters from ref. 42 was used to investigate the crater populations superposed on representative polar plains to infer their chronology and potential source regions. During the crater population analysis, possible secondary impact craters or non-impact pits were eliminated on the basis of their known characteristics and morphologies[59,60]. Empirical radial thickness models of impact ejecta materials (ref. 44 and references therein) were utilized to access the potential material provenance of representative polar plains.

We also characterized the surface optical albedo of the mapped polar plains terrains and compared them with those of typical landforms of the Moon (southern polar region, mare, light plains, and the global Moon) using the LOLA active laser measurements[25,61]. This unique albedo data was acquired by the LOLA laser instrument through measuring the laser energy at the 1064 nm wavelength reflected from lunar surface. As LOLA carries its own light source, it has the unique ability of mapping the surface reflectivity of lunar surface that are obscured from solar illumination. This is especially useful for the lunar polar regions where the solar altitude is always very low and large areas are within permanent shadows. The gridded LOLA albedo data used in this work was sampled from original point measurements at a resolution of 1 km/pixel for polar regions and 10 pixels/degree (about 3 km/pixel at the equator) for the global Moon. In addition, we investigated and compared the surface iron elemental abundance of various lunar terranes using the gridded FeO map (pixel size of 0.5 degree, or ~15 km at the equator) acquired by the Lunar Prospector gamma ray spectrometer (GRS)[27]. In particular, the FeO abundance map of lunar polar regions (1 km/pixel) derived from the Kaguya Spectral Profiler (SP) and LOLA data and calibrated to the Lunar Prospector GRS FeO map[26] was employed to characterize the surface chemical composition of the mapped polar plains, due to its significantly higher spatial resolution than Lunar Prospector GRS data. Optical albedo and iron content are key measurements for distinguishing typical lunar surface material types, as they are characterized by distinctly different optical and compositional signatures. Specifically, the LOLA albedo and WAC mosaic images are employed to search for impact craters with dark ejecta (dark-halo craters) on the mapped plains. We also use the Kaguya-SP and LRO-LOLA blended mineral abundance maps (1 km/pixel)[26] to characterize potential mafic-rich regolith in the polar plains. The presence of dark-halo craters and mafic-rich regolith are important criteria to distinguish whether these plains are underlain by mare volcanic deposits (cryptomeria)[62].

## Data availability

The LOLA topography and albedo, and Lunar Prospector gamma ray spectrometer data are archived at the NASA Planetary Data System Geosciences Node (https://pds-geosciences.wustl.edu/), the LROC WAC image mosaic is available at the LROC Archive (http://lroc.sese.asu.edu/), and the plains mapping data are accessible at Zenodo (https://doi.org/10.5281/zenodo.10799097).

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

## Acknowledgements
This work is supported by the National Natural Science Foundation of China (42241107, 12303064, 42372277), Pre-research Project on Civil Aerospace Technologies of CNSA (D020204), Science and Technology Development Fund, Macau (0012/2023/RIA1), State Key Laboratory of Lunar and Planetary Sciences of M.U.S.T (Macau FDCT grant No. SKL-LPS(MUST)–2021-2023), Young Scholars Program of Shandong University, Weihai (202207), Open Research Fund Program of LIESMARS, Wuhan University (Grant No. 22P02), and the Shandong Provincial Natural Science Foundation (ZR2023MD010).

## Author contributions
L.Q. designed the research, L.Q., L.X., J.C., and Y.Z. performed the data analysis, L.Q., L.X., and J.C. contributed to data visualization, and L.Q., J.W.H., L.X., J.C., Y.Z., B.L., and Z.L. contributed to the scientific interpretation and manuscript writing.

## Competing interests
The authors declare no competing interests.
