## [Peer Review File · Nature Communications]

Geological Evidence for Extensive Basin Ejecta as Plains Terrains in the Moon's South Polar RegionREVIEWER COMMENTS

Reviewer #1 (Remarks to the Author):

The manuscript "Geological Evidence for Extensive Basin Ejecta as Plains Terrains in the Moon's South Polar Region" presents an analysis of distinct smooth plains units across the South Pole region of the Moon, presenting a clear argument for potential modes of origin and sources of these features. The research presented in the manuscript is original, integrating across multiple datasets and using various models to develop a possible origin for the plains units found in the circumpolar region. In general, the data presented support the authors' hypotheses, and the authors do a good job of explaining their interpretations. The figures presented in the manuscript are adequate, but the authors may consider some revisions based on the comments below.

Line 68: The use of Figure 1 to make the point that the South Pole is distinct from the other areas of exploration, is not quite clear from this figure especially since the data aren't geologic maps, so it may be worth just saying that the area lacks the inferred volumes of volcanic material that have been sampled elsewhere?

Lines 68-77: It may be worth pointing out here that while the terrain is "very rugged" it's no different than other heavily cratered terrain on the Moon, like the FHT.

Lines 86 and 87: Here you compare the areas of high topography with the flat regions, it may be worth beginning to quantify what "rugged" and "very flat" are. The reason for this is to highlight that while the "rugged" areas are indeed higher topographically, they are no different than any other portion of the ancient crust. There's a mis-conception in the community that the South Pole is wildly different than the rest of the Moon (largely because of images like Figure 1a, which do a great job of showing the morphology, but also seem to suggest that the terrain is markedly different).

Line 82-85, 117: Figure 2: This figure is somewhat confusing, mixing shaded relief (hillside), slope, and albedo image data. I might suggest trying to directly compare Amundsen and either Aitken or Colombo but not both.

Line 127: I don't think you define it, but you should set what you mean by "South Polar region" as being areas poleward of 80 degrees south, just to be explicit.

Line 130 - You should explain why you don't catalogue plains units smaller than 1 km^2 .

Line 141: Is there a way to define "Smooth" or is that subjective?

Line 155-158: Are there any other reasons for the differences in the datasets used in the Meyer study vs. what is presented here? Are there differences in how units were defined?

Line 190-200: Given that the various datasets used for this comparison (LP derived FeO, LOLA albedo) have different spatial resolutions, are there any considerations for the cross-comparison of the values we should be aware of?

Line 244: Could you expand a little on why the Amundsen plains are not likely cyroptomare? Are Cryptomare units included in Figure 5?

Line 256: How many other craters did you examine for this comparison?

Line 272: Did you examine any units within Orientale basin, or just those outside of the basin? How similar/dissimilar is the Montes Rook Formation of Orientale?

Lines 315-339: Here you calculate possible ejecta thicknesses to fill Amundsen, are there any mechanisms for calculating the actual thickness of the deposits in Amundsen? It does not appear that there are any craters that excavate through the fill, but can you place a limit based on those that are inside the crater?

Line 350: It may be worth including a reference to figure S9 here.

Line 356: This may not be a necessary figure for the main text.

Line 348-387: It should be noted that the landing ellipse for Artemis may be much less than that of the Luna 25 mission.

Other notes: Are there any morphologies within the plains units that should be discussed? For example, in the plains units between de Gerlache and Ibn Bajja are appear to resemble wrinkle ridges?

Are there any plains units within the 13 Artemis III Candidate regions, this might be worth mentioning in your Discussion section.

Reviewer #2 (Remarks to the Author):

This manuscript presents new mapping and analysis of light plains in the south polar region of the Moon. The authors interpret the origin of these plains as distal impact ejecta, especially from the Schrodinger basin, and discuss the potential of these plains to provide safe landing sites for future landed exploration and the potential science advances that would come from investigating the plains.

Overall, the new mapping results provide a modest update to past work, and will be of interest to those examining the geology of the Moon's south pole. The interpretations regarding the origin of the plains are rather superficial, and do not substantially advance our understanding of their origin. Additional consideration should be given to the contributions of the many other basins that will have contributed to the origin of plains deposits in this region. Detailed comments follow.

Lines 68–75: The Moon's south pole is not more rugged than other highland regions, and the text here could perpetuate this myth that stems from the large shadows. "Very rugged" should be quantified or placed in context with other highland regions. Further, extensive flat terrain is not needed to ensure a safe landing for many upcoming missions. The last sentence here is redundant with earlier text in this section.

Line 82–85: The comparison with mare-filled equatorial craters is fine, but the light plains in Amundsen and other polar craters are also morphologically similar to many equatorial light plains that have previously been mapped. It is unclear why the comparison shown here is only to equatorial mare-filled craters. The introduction does not provide adequate context for the globally mapped light plains of, e.g., the Meyer et al. work that is cited later. The contrast stretch in Fig. 2c is also very low; a better stretch would highlight the clear albedo difference between the mare and surroundings.

Line 87: The manuscript makes liberal use of the word "very" – e.g., here, "very flat" plains, but this word is open to wide interpretation. How much flatter, and over what baselines?

Lines 89–98: I am not sure that all of these questions "remain to be investigated." This work provides some modest advances, but all of these questions have been addressed to some degree in past work. (For example, it is generally well accepted that none of these regions are basalt or cryptomare, but that they originate as ejecta similar to the Apollo 16 region.)

Line 111: add "(b)" before LOLA DEM. Was DEM defined earlier in the text?

Line 128: With plains ≥ 1 km², I think some of the data sets would not have sufficient resolution to really characterize the smallest areas?

Line 155: Did the scale at which Meyer et al. mapped features (1:300K) also contribute to the difference?

Figure 4: Image source? LROC?

Line 173–175: Is there a typo here? It seems that –7000 to –6700 m is not a wide range and should maybe be +6700 m based on Fig. S3a?

Lines 214–222: If the Schrodinger plains include distinctive features not shared with the other plains they are being grouped with, perhaps a Schrodinger is not an appropriate type example, and instead one of the others could be used. Also, there is no figure given that provides a good view of

the two types of plains and the morphologies that distinguish them, which is needed (Fig. 4a and 4c could potentially serve this purpose with additional info in the caption or arrows added to highlight key features).

Line 228: I'm surprised to see Shackleton mentioned here. Its floor is hummocky and not plains like. It also doesn't seem to be mapped as plains in Fig. 3.

Line 237: Is "flowable" a word? Fluid? Materials that can flow to reach an ~equipotential surface?

Line 249: The ballistic sedimentation process seems to include low-angle ejecta sweeping and churning up the surface, so describing this as "impact ejecta flows" is a bit of a simplification or could give readers the wrong mental image of how the plains may have been emplaced.

Line 265: "On the basis of these analog and process studies" – which studies? Alternatives – such as melt from the Amundsen impact event – are dismissed rather out-of-hand. This is a large impact crater and certainly part of the flat floor is due to impact melt. Maybe later impacts mixed/covered it, but surely some of the material is Amundsen melt. (Cryptomelt? Just kidding! No need to invent a term, but a concept similar to cryptomare.) This section is example of some of the over-simplification in this manuscript.

Line 286: Many references on impact melt that could be helpful to readers if included here.

Line 299: Typo - extra s on younger

Line 301: Craters included range from Eratoshenian to Nectarian – it does not seem appropriate to describe their ages as "young" and it is surprising that a Nectarian-age crater has primary melt-related morphology preserved. There are only nine plains of this type – a supplemental figure that shows the morphology of each would be helpful. (Also n=9, but 2 Eratosthenian + 5 Imbrian + 1 Nectarian = 8.)

Lines 307–309: The description here ignores mixing in of local materials when the distal ejecta is emplaced (see, e.g., Oberbeck work and application by Petro and Pieters). It is not simply a blanket of distal ejecta material, but thought to be a mix that includes varying portions of the local substrate. I think this is an important point to highlight for sample provenance.

Line 334 and surroundings: Schrodinger is likely a major source of light plains and one of the most recent significant contributors, but the way this is presented is very definitive and ignores other likely contributors to the plains in the region. There appears to be light plains/ejecta covering some of the mare within Schrodinger so some large impact event affected the region post-Schrodinger (Orientale?), Orientale definitely affected the area to some extent, and the many basins around the area must have also contributed. This should be described to acknowledge the complexity.

Lines 350–351: The ages of Schrodinger and Amundsen plains are similar, but it seems a stretch to call them nearly identical. I'm a skeptic of the small model age error values, but 3.8 ± 0.02 Ga does

not overlap 3.9 ± 0.01 Ga, and this issue is not discussed. If those values are to be believed, Schrodinger is older than Amundsen and could not have contributed at all to the plains in the interior. Further, the Schrodinger count area is complex with many secondaries. There is a ghost crater (filled in with light plains, see screen grab in pdf) that suggests the area has been resurfaced after some amount of time during which that crater formed post-formation of Schrodinger. If this is the case, then the Schrodinger impact event is even older than the surface age this count area would indicate.

General comment on “Results” section: includes lots of interpretation that should really be in the “Discussion” section.

Line 365: popular targets for future/planned exploration (none of these missions to the south polar region have yet happened)

Line 366: Large-scale rugged topography compared to what – the mare? ~Similar to highlands.

Line 396: “areally extensive areas” – remove areally

Line 412: The sentence starting with “In addition” is a non sequitur from what is discussed in the rest of the paragraph.

Line 421: “Adequate solar energy” will be mission dependent.

Line 428 (and abstract and line 431 and 438): I think describing the ejecta as “foreign” and having “contaminated” is the wrong framing. Ejecta materials from possibly distant impact craters/basins, with similar highland composition. This is true basically anywhere in the lunar highlands. The final paragraph here again discusses local material being “buried” but local material is thought to be incorporated into the deposit via ballistic sedimentation.

Lines 467–471: I believe this red-green-blue method is similar to Meyer et al., if so should be referenced.

Line 505: Lunar Prospector data resolution is altitude dependent, 0.5 degrees must be equatorial degrees because the resolution is more like 30 km not 2.6 km. Thus using LP FeO data is likely not really appropriate for distinguishing among the compositions of small plains deposits.

Response to Reviewers:
The Nature Portfolio Journal *Nature Communications*
Ref.: Manuscript #NCOMMS-23-55796
“Geological Evidence for Extensive Basin Ejecta as Plains Terrains in the Moon’s South Polar Region”

(Comments in normal type, responses in ***bold italics***)

REVIEWER COMMENTS

Reviewer #1 (Remarks to the Author):

The manuscript "Geological Evidence for Extensive Basin Ejecta as Plains Terrains in the Moon’s South Polar Region" presents and analysis of distinct smooth plains units across the South Pole region of the Moon, presenting clear argument for potential modes of origin and sources of these features. The research presented in the manuscript is original, integrating across multiple datasets and using various models to develop a possible origin for the plains units found in the circumpolar region. In general the data presented support the authors hypotheses, and the authors do a good job of explaining their interpretations. The figures presented in the manuscript are adequate, but the authors may consider some revisions based on the comments below.

Response: We sincerely thank reviewer #1 for the very supportive and helpful comments. All these comments are responded to below.

Line 68: The use of Figure 1 to make the point that the South Pole is distinct from the other areas of exploration, is not quite clear from this figure especially since the data aren't geologic maps, so it may be worth just saying that the area lacks the inferred volumes of volcanic material that have been sampled elsewhere?

Response: Yes, the reviewer is correct, and we have clarified this in the revised manuscript.

Lines 68-77: It may be worth pointing out here that while the terrain is "very rugged" its no different than other heavily cratered terrain on the Moon, like the FHT.

Response: Yes, we agree and have incorporated this point.

Lines 86 and 87: Here you compare the areas of high topography with the flat regions, it may be worth beginning to quantify what rugged and "very flat" are. The reason for this is to highlight that while the "rugged" areas are indeed higher topographically, they are no different than any other portion of the ancient crust. There's a mis-conception in the community that the south pole is wildly different than the rest of the Moon (largely because of images like Figure 1a, which do a great job of showing the morphology, but also seem to suggest that the terrain is markedly different.

Response: We have quantified the topographic slopes of the mapped plains and the entire southern polar region using both median values (slope: ~3° vs. ~8°; roughness at 660 m baseline: ~7 m vs. 25 m) and histogram plots (Supplementary Fig. 1) in the revised manuscript.

Yes, we agree with reviewer that the southern polar region is topographically similar to “any other portion of the ancient crust”, as we have addressed in the response to the “Lines 68-77” comments above.

Line 82-85, 117: Figure 2: This figure is somewhat confusing, mixing shaded relief (hillside), slope, and albedo image data. I might suggest trying to directly compare Amundsen and either Aitken or Colombo but not both.

Response: Thanks for this very helpful suggestion.

We have labelled the location and source data for the Fig. 2 maps, and removed the Aitken map in the revised manuscript. We also added a map for the Plummer crater that contains light plains on the floor, according to one of the comments (Line 82–85) from Reviewer #2.

Line 127: I don't think you define it, but you should set what you mean by "South Polar region" as being areas poleward of 80 degrees south, just to be explicit.

Response: We have defined the studied area (southward of 80° latitude) in the last paragraph of

the Introduction section of the revised manuscript.

Line 130 - You should explain why you don't catalogue plains units smaller than 1 km².

Response: We have added the explanation that the minimal size of the mapped plains units is limited by the 20–1000 m/pixel spatial resolution of the various data sets used in this work (see revised Methods).

Line 141: Is there a way to define "Smooth" or is that subjective?

Response: We have referred to the RMS height roughness calculated in this work to quantify the smoothness (topographical roughness typically < 10 m at 660 m baseline) in the revised manuscript.

Line 155-158: Are there any other reasons for the differences in the datasets used in the Meyer study vs. what is presented here? Are there differences in how units were defined?

Response: We addressed the point that the difference in the mapping scale (1:300K vs. various scales from 1:200K to 400K) was additionally one of the reasons, but the major reason remains to the new LOLA data sets with much higher resolution (20 m/pixel) and better spatial coverage employed in this work, compared with the 100 m/pixel LROC WAC data used by Meyer et al., with considerable shadowed areas.

The definition of plains units in this work is similar to that of the Meyer study, and we have specifically added the citation to the Meyer study in the Methods section.

Line 190-200: Given that the various datasets used for this comparison (LP derived FeO, LOLA albedo) have different spatial resolutions, are there any considerations for the cross-comparison of the values we should be aware of?

Response: Thanks for this very constructive suggestion.

We have reanalyzed the FeO abundance of the polar region using the new FeO abundance map of lunar polar regions derived from the Kaguya Spectral Profiler (SP) and LOLA data and calibrated to the Lunar Prospector GRS FeO map (Lemelin et al., 2022), which has the same 1 km/pixel spatial resolution of the LOLA albedo map.

In addition, we analyzed the FeO abundance of non-polar regions (the global Moon, maria, and light plains) using the LP GRS data, which indeed has a much lower resolution. Cross-comparison between the two FeO datasets (cf. Fig 5b and Supplementary Fig. 4) indeed reveals a small discrepancy in the FeO abundance (e.g., 7.2% vs. 6.5% median value for the southern polar region). We have discussed this issue in the revised manuscript (Line 207-212).

Line 244: Could you expand a little on why the Amundsen plains are not likely cryptomare? Are Cryptomare units included in Figure 5?

Response: Thanks for this very constructive comment concerning cryptomeria as a candidate for the nature and origin of the Amundsen plains. We have addressed this concern through additional data analyses and geological deductions in the revised manuscript:

- (1) We systematically search for impact craters with dark ejecta (dark-halo craters) that might indicate the presence of cryptomaria on the mapped polar plains using both LOLA albedo and WAC mosaic images;***
- (2) We employ mineral abundance maps derived from the Kaguya Spectral Profiler (SP) and LOLA data to characterize potential mafic-rich regolith that might indicate the presence of cryptomaria in the polar plains;***
- (3) On the basis of the presence/absence of dark-halo crater and mafic-rich regolith, we determine whether the evidence suggests that the mapped polar plains are underlain by mare volcanic deposits (cryptomaria);***

Our detailed investigations do not observe any dark-halo impact craters (consistent with Izquierdo et al., 2024, JGR-Planets, 129, e2023JE007867) or mafic-rich regolith signature in the plains terrains that we studied, providing no evidence of mare or cryptomaria for the origin of these plains. This result is also consistent with the deficiency of volcanic deposits detected in the polar regions of the Moon (e.g., Krasilnikov et al., 2023, Icarus, 394, 115422; Broquet & Andrews-Hanna, 2024, Icarus, 411, 115954; Head et al., 2023, Reviews in Mineralogy & Geochemistry, 89, 453-507).

Line 256: How many other craters did you examine for this comparison?

Response: Thanks for this very helpful comment. We have examined all impact craters (n=187 according to the catalogue of Head et al. (2010, Science, 329, 1504-1507)) with sizes (90–110 km diameter) similar to Amundsen crater (103 km diameter). We have added this description in the revised manuscript.

Line 272: Did you examine any units within Orientale basin, or just those outside of the basin? How similar/dissimilar is the Montes Rook Formation of Orientale?

Response: Thanks for this very insightful comment. We only investigated the units outside of the Orientale basin in the initial manuscript.

We have examined the Maunder (inside the ~480 km diameter Inner Rook Ring) and Montes Rook (between the ~930 km diameter Cordillera Ring and the ~620 km diameter Outer Rook Ring) Formations of Orientale in the revision, and indeed found that many units are morphologically similar with the Schrödinger-type plains terrains mapped in our study: predominately occurring in topographically low regions, and consisting mainly of flat-lying materials and km-scale blocks.

This new comparison provides additional important insights into the nature and emplacement mechanism of the Schrödinger-type polar plains. These units of the Maunder and Montes Rook Formations have been commonly interpreted as clast-rich melt materials deposited at or just beyond the extent of the transient cavity for Orientale (Head, 1974; Scott et al. 1977), consistent with our local impact melt interpretation for these Schrödinger-type plains. We have incorporated these new analyses and a new map of units in the Montes Rook Formation (Fig. 4e) in the revised manuscript.

Lines 315-339: Here you calculate possible ejecta thicknesses to fill Amundsen, are there any mechanisms for calculating the actual thickness of the deposits in Amundsen? It does not appear that there are any craters that excavate through the fill, but can you place a limit based on those that are inside the crater?

Response: Thanks for this very constructive comment. We have used the excavation depths (0.084 times crater rim-to-rim diameter) of the superposed impact craters (up to ~2.9 km in diameter) to constrain the upper limit (~250 m) of the thickness of Amundsen floor plain materials.

Line 350: It may be worth including a reference to figure S9 here.

Response: Yes, incorporated.

Line 356: This may not be a necessary figure for the main text.

Response: We feel that these maps are essential for the analog studies to interpret the formation mechanism of the mapped plains terrains. To make such comparison more apparent and reduce the paper length, we have merged it into Fig. 4 in the revised manuscript.

Line 348-387: It should be noted that the landing ellipse for Artemis may be much less than that of the Luna 25 mission.

Response: Thanks for this very useful suggestion. We have added a note that the landing ellipse for NASA's Artemis program is probably much smaller, at a scale of ~100 m (e.g., Chavers et al., 2020, 2020 IEEE Aerospace Conference, 1-9), in the revised manuscript.

Other notes: Are there any morphologies within the plains units that should be discussed? For example, in the plains units between de Gerlache and Ibn Bajja are appear to resemble wrinkle ridges?

Response: Thanks for this very helpful comment. The wrinkle ridges in the lunar south polar region definitely deserve dedicated studies, like this one between de Gerlache and Ibn Bajja, as many lunar wrinkle ridges are usually interpreted to be related to underlying layered strata, especially layered mare basalts. We have examined the distribution of wrinkle ridges using various data sets including images, topography, and spectroscopy, and found no evidence of correlation with buried mare basalts, consistent with the geological mapping of Krasilnikov et al. (2023, Icarus, 394, 115422).

Are there any plains units within the 13 Artemis III Candidate regions, this might be worth mentioning in your Discussion section.

Response: *Thanks for this very constructive comment.*

We examined the 13 Artemis III candidate regions (<https://www.nasa.gov/news-release/nasa-identifies-candidate-regions-for-landing-next-americans-on-moon/>) and found that 8 of them contained the plains terrains mapped in this work, though these plains are typically very small in size (several km) and occurs at the marginal low-lying portion of the candidate regions. We have added these discussions and a new map (Supplementary Fig. 13) of the Peak Near Shackleton region in the supplement to illustrate this observation.

Reviewer #2 (Remarks to the Author):

This manuscript presents new mapping and analysis of light plains in the south polar region of the Moon. The authors interpret the origin of these plains as distal impact ejecta, especially from the Schrodinger basin, and discuss the potential of these plains to provide safe landing sites for future landed exploration and the potential science advances that would come from investigating the plains.

Overall, the new mapping results provide a modest update to past work, and will be of interest to those examining the geology of the Moon's south pole. The interpretations regarding the origin of the plains are rather superficial, and do not substantially advance our understanding of their origin. Additional consideration should be given to the contributions of the many other basins that will have contributed to the origin of plains deposits in this region. Detailed comments follow.

Response: *We sincerely acknowledge Reviewer #2 for the very supportive and constructive comments. All these comments are responded to below.*

Specifically, as we have responded to above, we have thoroughly addressed the concerns concerning the interpretations regarding the origin of the plains, through additional data analyses and geological deductions in the revised manuscript:

- (1) We systematically search for impact craters with dark ejecta (dark-halo craters) that might indicate the presence of cryptomaria on the mapped polar plains using both LOLA albedo and WAC mosaic images;**
- (2) We employ mineral abundance maps derived from the Kaguya Spectral Profiler (SP) and LOLA data to characterize potential mafic-rich regolith that might indicate the presence of cryptomaria in the polar plains;**
- (3) On the basis of the presence/absence of dark-halo crater and mafic-rich regolith, we determine whether the evidence suggests that the mapped polar plains are underlain by mare volcanic deposits (cryptomaria);**
- (4) We explored alternative scenarios for the nature and origin of the studied plains terrains, including both exposed (maria) and obscured (cryptomaria) basaltic volcanism;**
- (5) We also discussed the challenges and limitations for identifying surface albedo variations in the polar regions using typical conventional optical images due to considerable shadows and typically very low Sun illumination.**

Our detailed investigations do not observe any dark-halo impact craters (consistent with Izquierdo et al., 2024, JGR-Planets, 129, e2023JE007867) or mafic-rich regolith signature in the plains terrains that we studied, providing no evidence of mare or cryptomaria for the origin of these plains. This result is also consistent with the deficiency of volcanic deposits detected in the polar regions of the Moon (e.g., Krasilnikov et al., 2023, Icarus, 394, 115422; Broquet & Andrews-Hanna, 2024, Icarus, 411, 115954; Head et al., 2023, Reviews in Mineralogy & Geochemistry, 89, 453-507).

Lines 68–75: The Moon's south pole is not more rugged than other highland regions, and the text here could perpetuate this myth that stems from the large shadows. "Very rugged" should be quantified or placed in context with other highland regions. Further, extensive flat terrain is not needed to ensure a safe landing for many upcoming missions. The last sentence here is redundant with earlier text in this section.

Response: *Yes, we agree with the reviewer that "the Moon's south pole is not more rugged than other highland regions", and we have incorporated this in the revised manuscript.*

We have also quantified the “ruggedness” of the Moon’s south pole here and at multiple locations in the revised manuscript, and reworded the last sentence.

Line 82–85: The comparison with mare-filled equatorial craters is fine, but the light plains in Amundsen and other polar craters are also morphologically similar to many equatorial light plains that have previously been mapped. It is unclear why the comparison shown here is only to equatorial mare-filled craters. The introduction does not provide adequate context for the globally mapped light plains of, e.g., the Meyer et al. work that is cited later. The contrast stretch in Fig. 2c is also very low; a better stretch would highlight the clear albedo difference between the mare and surroundings.

Response: Thanks for these very constructive suggestions.

We have (1) added a map for an equatorial crater (Plummer, Fig. 2d) that was filled with light plains mapped previously (Meyer et al., 2020), (2) added adequate context for the globally mapped light plains (e.g., Wilhelms et al., 1979; Meyer et al., 2020) in the introduction section, and (3) remapped Fig. 2c using WAC morphological mosaic (other than the low-sun mosaic in the initial manuscript) for better stretching the albedo difference.

Line 87: The manuscript makes liberal use of the word “very” – e.g., here, “very flat” plains, but this word is open to wide interpretation. How much flatter, and over what baselines?

Response: We have quantified these descriptions here and in multiple locations in the revised manuscript.

Lines 89–98: I am not sure that all of these questions “remain to be investigated.” This work provides some modest advances, but all of these questions have been addressed to some degree in past work. (For example, it is generally well accepted that none of these regions are basalt or cryptomare, but that they originate as ejecta similar to the Apollo 16 region.)

Response: Thanks for this very helpful suggestion. We have reworded it to “remain to be further investigated” to make it more accurate.

Also, as reflected by one of the Reviewer #1’s comments (Line 244), we are not sure “that they are not cryptomaria” is “generally accepted”. On the basis of discussions at the recent Lunar and Planetary Science Conference 55, we believe that some people in the community are still not convinced, and thus we prefer to leave this as is.

Line 111: add “(b)” before LOLA DEM. Was DEM defined earlier in the text?

Response: Thanks. As the term DEM was only used once in the manuscript, we have changed it to “gridded” in the revised manuscript.

Line 128: With plains ≥ 1 km², I think some of the data sets would not have sufficient resolution to really characterize the smallest areas?

Response: Thanks for this important comment.

We have reanalyzed the FeO abundance of the polar region using the new Kaguya-SP and LRO-LOLA blended FeO map (1 km/pixel, Lemelin et al., 2022), rather than the 0.5 degree/pixel LP GRS data used in the initial manuscript. Now, the various data sets used in this work have pixel sizes ranging from 20 m to 1 km, sufficient to resolve all plains ≥ 1 km².

Line 155: Did the scale at which Meyer et al. mapped features (1:300K) also contribute to the difference?

Response: Yes, we agree. We have added this explanation in the revised manuscript.

Figure 4: Image source? LROC?

Response: We have added descriptions of the image sources (LOLA shaded relief and LROC WAC mosaic) in each panel and the figure caption.

Line 173–175: Is there a typo here? It seems that –7000 to –6700 m is not a wide range and should maybe be +6700 m based on Fig. S3a?

Response: Thanks for the careful review. Yes, it should be +6700 m here. We have fixed this error.

Lines 214–222: If the Schrodinger plains include distinctive features not shared with the other plains they are being grouped with, perhaps a Schrodinger is not an appropriate type example, and instead

one of the others could be used. Also, there is no figure given that provides a good view of the two types of plains and the morphologies that distinguish them, which is needed (Fig. 4a and 4c could potentially serve this purpose with additional info in the caption or arrows added to highlight key features).

Response: Yes, we agree with the reviewer on the distinctive features on the Schrödinger plains. However, the other mound-bearing plains (~1–102 km²) are over one order of magnitude smaller than the Schrödinger plains (~8576 km² mapped in a small part of the Schrödinger floor). The other plains typically occupy only a very small portion of the floor of the host crater (Supplementary Figs. 5 and 6) and their morphological characteristics are not as apparent and representative as the Schrodinger floor plains. Therefore, we prefer to retain Schrödinger as a type example.

We have added descriptions and arrows in Figs. 4a and 4c to highlight key features of these plains.

Line 228: I'm surprised to see Shackleton mentioned here. Its floor is hummocky and not plains like. It also doesn't seem to be mapped as plains in Fig. 3.

Response: Yes, the reviewer is correct that the Shackleton floor is mainly hummocky, but two very small areas (both ~1.5 km²) are relatively flat and mapped as plains (Supplementary Fig. 6g; also see figure below).

The Shackleton floor plains have actually been mapped (color patches) in Fig. 3, but it is too small to be seen in the initial manuscript. We have re-made the Fig. 3 map (add outlines) to make it more visible.

Line 237: Is “flowable” a word? Fluid? Materials that can flow to reach an ~equipotential surface?

Response: Thanks. We have replaced it with “fluidized”.

Line 249: The ballistic sedimentation process seems to include low-angle ejecta sweeping and churning up the surface, so describing this as “impact ejecta flows” is a bit of a simplification or could give readers the wrong mental image of how the plains may have been emplaced.

Response: Thanks for this very insightful comment.

We agree that many ejecta materials would excavate and mix with preexisting terrain materials when they re-impacted the lunar surface. The mixtures of distant and local debris would move laterally and form the plains terrains (e.g., Oberbeck, 1975, Reviews of Geophysics, 13, 337-

362). We have re-worded it to “impact debris flows consisting of pre-impact substrate and ejecta materials” to acknowledge the complexity.

Line 265: “On the basis of these analog and process studies” – which studies? Alternatives – such as melt from the Amundsen impact event – are dismissed rather out-of-hand. This is a large impact crater and certainly part of the flat floor is due to impact melt. Maybe later impacts mixed/covered it, but surely some of the material is Amundsen melt. (Cryptomelt? Just kidding! No need to invent a term, but a concept similar to cryptomare.) This section is example of some of the over-simplification in this manuscript.

Response: We meant the studies described earlier in this paragraph: comparison of the morphologies of Amundsen-type plains terrains (Figs. 4a,b) and light plains radial to the Orientale basin (Fig. 4d), and the related geological deductions.

Yes, we agree with reviewer on the underlying melt materials, and we have added such discussions in the revised manuscript.

Line 286: Many references on impact melt that could be helpful to readers if included here.

Response: Yes, we agree, and we have added the Osinski et al. (2011, EPSL, 310, 167-181) reference. Due to the reference number limit of the journal, we cannot add more references.

Line 299: Typo - extra s on younger

Response: fixed.

Line 301: Craters included range from Eratoshenian to Nectarian – it does not seem appropriate to describe their ages as “young” and it is surprising that a Nectarian-age crater has primary melt-related morphology preserved. There are only nine plains of this type – a supplemental figure that shows the morphology of each would be helpful. (Also n=9, but 2 Eratosthenian + 5 Imbrian + 1 Nectarian = 8.)

Response: Thanks for the very careful review.

We have (1) added a supplemental figure (Supplementary Fig. 6) that shows the morphology of all the 9 Schrödinger-type plains, and (2) re-checked our result and fixed the count error: 2 Eratosthenian + 5 Imbrian + 2 Nectarian = 9, in the revised manuscript.

We addressed the point that the host craters of Schrödinger-type plains are relatively younger than those of Amundsen-type plains (Nectarian and Pre-Nectarian).

The two Nectarian-age craters are Idel'son (57 km diameter, centered at 81.2°S, 112.7°E; Supplementary Fig. 6f) and one unnamed crater between Casatus D and Newton E (19 km, 78.6°S, 39.9°W; Supplementary Fig. 6a). We address the point that the stratigraphy of these craters is from the geological maps of Fortezzo et al. (2020) and Krasilnikov et al. (2023a), and these stratigraphical assignments are not of problem (e.g., McEwen et al., 1993, JGR, 98(E9), 17207–17231; Neukum & König, 1976, LPSC; Xie et al., JGR-Planets, 125, e2019JE006112; Wang et al., 2020, JGR-Planets, 125, e2019JE006091). For instance, the unnamed 19-km-diameter crater has relatively pristine appearances on both LOLA shaded relief and WAC image maps, suggesting that it is probably younger than the assigned Nectarian age.

We agree with the reviewers on the preservation of melt-related morphology in Nectarian-aged craters, but impact melt morphologies have been previously mapped in Nectarian-aged craters (e.g., the Crisium basin; Spudis & Sliz, 2017, GRL, 44, 1260–1265; Runyon et al., 2020, JGR-Planets, 125, e2019JE006024).

We have addressed these issues in the revised manuscript.

Lines 307–309: The description here ignores mixing in of local materials when the distal ejecta is emplaced (see, e.g., Oberbeck work and application by Petro and Pieters). It is not simply a blanket of distal ejecta material, but thought to be a mix that includes varying portions of the local substrate. I think this is an important point to highlight for sample provenance.

Response: Yes, we totally agree with the reviewer on this very important point. We have addressed the mixing of local materials and implications for sample provenance throughout the revised manuscript.

Line 334 and surroundings: Schrodinger is likely a major source of light plains and one of the most recent significant contributors, but the way this is presented is very definitive and ignores other likely contributors to the plains in the region. There appears to be light plains/ejecta covering some of the mare within Schrodinger so some large impact event affected the region post-Schrodinger (Orientale?), Orientale definitely affected the area to some extent, and the many basins around the area must have also contributed. This should be described to acknowledge the complexity.

Response: Thanks for this very constructive comment.

Yes, we agree with the reviewer that additional basins should have also contributed to the mapped plains materials, especially the Orientale basin, as it is younger and its ejecta should have been superposed on and mixed with the Schrödinger ejecta.

We have acknowledged the complexity in the revised manuscript.

Lines 350–351: The ages of Schrodinger and Amundsen plains are similar, but it seems a stretch to call them nearly identical. I'm a skeptic of the small model age error values, but 3.8 ± 0.02 Ga does not overlap 3.9 ± 0.01 Ga, and this issue is not discussed. If those values are to be believed, Schrodinger is older than Amundsen and could not have contributed at all to the plains in the interior. Further, the Schrodinger count area is complex with many secondaries. There is a ghost crater (filled in with light plains, see screen grab in pdf) that suggests the area has been resurfaced after some amount of time during which that crater formed post-formation of Schrodinger. If this is the case, then the Schrodinger impact event is even older than the surface age this count area would indicate.

Response: Thanks for these very careful and constructive comments.

We agree with the reviewer on the issues concerning the small model age error values. We have addressed that the difference between the model ages of Schrodinger and Amundsen plains are well within the uncertainties of the crater population dating technique (e.g., Fassett, 2016 and many references therein) in the revised manuscript.

We have checked the detailed morphology of the potential “ghost” crater using high-resolution Kaguya TC (10 m pixel size, upper figure) and LROC NAC (~1 m pixel size, middle figure) images, and did not observe the embayment of the crater rim by the adjacent materials (an example of a ghost crater at 40.39°N, 51.21°W is shown in the lower figure). We attribute its flat floor to the highly degradation morphology of this crater.

General comment on “Results” section: includes lots of interpretation that should really be in the “Discussion” section.

Response: Thanks for this very helpful suggestion, and we have moved some interpretations to the “Discussion” section.

Line 365: popular targets for future/planned exploration (none of these missions to the south polar region have yet happened)

Response: Yes, and we have reworded this.

Line 366: Large-scale rugged topography compared to what – the mare? ~Similar to highlands.

Response: Yes, it is compared with mare plains that many previous missions landed on, and we have reworded this.

Line 396: “areally extensive areas” – remove areally

Response: Fixed.

Line 412: The sentence starting with “In addition” is a non sequitur from what is discussed in the rest of the paragraph.

Response: We have reworded it.

Line 421: “Adequate solar energy” will be mission dependent.

Response: Yes, and we have reworded it.

Line 428 (and abstract and line 431 and 438): I think describing the ejecta as “foreign” and having “contaminated” is the wrong framing. Ejecta materials from possibly distant impact craters/basins, with similar highland composition. This is true basically anywhere in the lunar highlands. The final paragraph here again discusses local material being “buried” but local material is thought to be incorporated into the deposit via ballistic sedimentation.

Response: Thanks for this very helpful suggestion. We have reworded the manuscript using “distant” instead of “foreign” and “contributed” instead of “contaminated”.

Lines 467–471: I believe this red-green-blue method is similar to Meyer et al., if so should be

referenced.

Response: Yes, the reviewer is correct. We have added the citation to the Meyer study in the Methods section.

Line 505: Lunar Prospector data resolution is altitude dependent, 0.5 degrees must be equatorial degrees because the resolution is more like 30 km not 2.6 km. Thus using LP FeO data is likely not really appropriate for distinguishing among the compositions of small plains deposits.

Response: Yes, we agree and have changed the description of the LP FeO data resolution to "pixel size of 0.5 degree, or ~15 km at the equator".

We have reanalyzed the FeO abundance of the polar region using the new Kaguya-SP and LRO-LOLA blended FeO map (1 km/pixel, Lemelin et al., 2022), other than the 0.5 degree/pixel LP GRS data used in the initial manuscript. The new FeO data should have sufficient resolution to resolve all plains ≥ 1 km² mapped in this work.

REVIEWERS' COMMENTS

Reviewer #2 (Remarks to the Author):

The authors have made important updates that improved the manuscript.

I think the authors may have missed updating the abstract - it still talks about "ejecta flows" and the sentence about "the entire lunar south polar region should have been contributed by distant basin materials" should be revised to better match the rest of the text.

Line 249 - suggest changing to plural - Origins and Source Regions

Line 342: update to "What are the source regions of the distal materials?" since the sentence before refers to local and distal material.

Line 358-360: How would you tell if a superposed crater excavated Amundsen crater floor vs. superposed plains? Compositions are likely very similar, I think it is not possible to make this claim. Instead you could note that if the light plains are as thick as predicted from ejecta models, none of the superposed craters are large enough to have excavated through them (if that is the case).

Line ~370: Another check of the Schrödinger idea would be to see if the abundance of light plains also follows this pattern moving away from Schrödinger in another direction. I understand the authors only mapped the polar area, but this should also be observed in the Meyer et al map.

Line 396: I suggest changing from "as the major source" to "as a major source". Picky, but allows for other major sources like Orientale.

Line 411: candidate landing regions, rather than sites.

Response to Reviewers:
The Nature Portfolio Journal *Nature Communications*
Ref.: Manuscript #NCOMMS-23-55796A
“Geological Evidence for Extensive Basin Ejecta as Plains Terrains in the Moon’s South Polar Region”

(Comments in normal type, responses in ***bold italics***)

REVIEWERS' COMMENTS

Reviewer #2 (Remarks to the Author):

The authors have made important updates that improved the manuscript.

I think the authors may have missed updating the abstract - it still talks about "ejecta flows" and the sentence about "the entire lunar south polar region should have been contributed by distant basin materials" should be revised to better match the rest of the text.

Response: Thanks for these very careful reviews.

We have changed "ejecta flows" to "debris flows", and "... should have been ..." to "probably have been" to better match the rest of the text (e.g., Lines 273 and 472 in the manuscript).

Line 249 - suggest changing to plural - Origins and Source Regions

Line 342: update to "What are the source regions of the distal materials?" since the sentence before refers to local and distal material.

Response: Yes, we have incorporated these two revisions.

Line 358-360: How would you tell if a superposed crater excavated Amundsen crater floor vs. superposed plains? Compositions are likely very similar, I think it is not possible to make this claim. Instead you could note that if the light plains are as thick as predicted from ejecta models, none of the superposed craters are large enough to have excavated through them (if that is the case).

Response: Yes, we agree that the compositions of the crater floor and superposed plains should be very similar. While we think that the primitive crater floor substrates are probably more cohesive than surface fine-grained debris flow materials. Impacts into these layered targets could form craters with unique morphologies, e.g., crater floors of concentric features. We have added this explanation in the revised manuscript.

Line ~370: Another check of the Schrödinger idea would be to see if the abundance of light plains also follows this pattern moving away from Schrödinger in another direction. I understand the authors only mapped the polar area, but this should also be observed in the Meyer et al map.

Response: Thanks for this very constructive suggestion.

We have added the suggested analysis and indeed found similar patterns, though the areal abundance of light plains stars to drop at locations relatively closer to the basin rim (~3 radii), likely due to the coverage of some plains by mare deposits (Supplementary Figs. 13 and 14). We have incorporated these analyses and interpretations in the revised manuscript.

Line 396: I suggest changing from "as the major source" to "as a major source". Picky, but allows for other major sources like Orientale.

Line 411: candidate landing regions, rather than sites.

Response: Yes, we have incorporated these two revisions.